# Dynamic Release of Solutes from Roof Bitumen Sheets Used for Rainwater Harvesting

**Uri Nachshon** [1,*] , **Meni Ben-Hur** [1,2] , **Daniel Kurtzman** [1], **Roee Katzir** [1], **Lior Netzer** [1,2,3], **Guy Gusser** [3] **and Yakov Livshitz** [3]

1    Institute for Soil, Water and Environmental Sciences, ARO, Volcani Research Center, Rishon Le-Tsiyon 7505101, Israel; meni@volcani.agri.gov.il (M.B.-H.); daniel@volcani.agri.gov.il (D.K.); roeek@volcani.agri.gov.il (R.K.); Liorn@water.gov.il (L.N.)
2    Department of Geography and Environmental Development, Ben Gurion University of the Negev, Beer Sheva 8410501, Israel
3    Israeli Hydrological Service, Israeli Water Authority, Jerusalem 9195021, Israel; GuyG@water.gov.il (G.G.); YakovL20@water.gov.il (Y.L.)
\*    Correspondence: urina@volcani.agri.gov.il

**Abstract:** Bitumen waterproof sheets are widely used to seal building roofs. Previous works have focused on the mechanical-physical properties of bitumen sheets, as well as their aging and degradation processes, and their impact on sealing properties of the buildings. Due to a growing need over recent years to use rooftops in urban environments for rainwater harvesting purposes, it is highly important to better characterize the quality of the harvested water from the bitumen covered roofs, and to shed more light on the impact of bitumen degradation processes on the release of various components to the harvested roof water. In the present study, the extracted organic and inorganic solutes from bitumen-covered roofs by water flow on the bitumen sheets were examined through a series of experiments, including measurements from the roofs of buildings in the center of Israel during the winter of 2019–2020. The results indicated high levels of organic and inorganic solute loads in the roof water during the first flush of the first rain of the winter, with maximal electric conductivity readings at the order of 4 dS/m. However, it was shown that following the first flush, a ~20 mm of cumulative rainfall was sufficient to wash off all the summers' accumulated solutes from the roof. After this solute flushing of the roof, harvested rainwater along the winter was of good quality, with electric conductivity readings in the range of 0.04–0.85 dS/m. Moreover, it was shown that bitumen sheets which were exposed to direct sun radiation emitted greater loads of solutes, likely a result of elevated aging and degradation processes. The findings of the present research point to the need to find efficient ways to isolate roof bitumen sheets from direct sun radiation and to design rainwater harvesting systems that will not collect the water drained from the first flush.

**Keywords:** rainwater harvesting; water quality; bitumen sheets; first flush

## 1. Introduction

### 1.1. Hydraulic Sensitivity of the Urban Environment

Recent years, including the summer of 2021, have shown the hazardous and immediate effect that global climate change has on our society and the natural environment [1,2]. It is well known that climate change is already affecting our planet, and a persistent increase in average temperatures is being observed, alongside increased occurrences of climatic extremes, such as droughts, extreme rain events, and fatal floods [3,4].

Under the conditions of the global change, urban areas are becoming a sensitive environment that is highly affected by the changing climate, with many aspects associated with the hydrological cycle. Cities have a great impact on the hydrology of their region as the consumption of good quality water is high, and the dense population produces high volumes of sewage, increasing the risk of natural water resource contamination.

Moreover, in the urban environment, there is a major reduction of rainwater infiltration and groundwater recharge due to the existence of large impervious areas and changes in the pattern of surface runoff flows [5]. As a result, during rain events, large volumes of water that, under natural conditions, could have enriched surface water bodies and below-ground aquifers, are being diverted into the municipal drainage system (MDS) and lost. As already seen in many places worldwide, these hydrological changes result in high peak flows and large runoff volumes during rain events [1,4], which increase the risk of economic, environmental, and life-threatening hazards, such as city floods and the overburden of the MDS [6].

The world's population is becoming concentrated in the cities, with more than 50% of the total world's population and more than 75% of the population in North America, Europe, and Oceania living in cities [7,8]. Consequently, there is a growing need to move toward the goal of efficient and appropriate use and management of natural water resources in urban environments, mainly in arid and semi-arid climates [9]. For this purpose, one of the practices that has gained popularity over recent years in urban environments is the use of rainwater harvesting (RWH). RWH, which is the collection, storage, transport, and use of rain water for different purposes [10], has been discussed in the scientific literature and applied in several places worldwide [6,9–15].

### 1.2. Rain Water Harvesting (RWH) and Water Quality

In the past, RWH consisted of rainwater collection systems where the harvested water was usually stored in underground cisterns or used directly for irrigation [6,16,17]. Nowadays, RWH, at the single building scale, consists of rainwater collection from large surfaces, mainly rooftops [18]. The collected water can be stored in below- or above-ground reservoirs [9,19,20]. The collected water can also be injected into the subsurface or used to recharge groundwater via designated infiltration basins and wells [6,12,20–22]. The quality of the harvested water, which is mainly affected by the quality and state of the water collection surfaces and delivery systems, determines whether the water can be used for groundwater recharge (directly to the aquifer or infiltrating through the unsaturated zone), direct drinking, domestic uses (car wash for example), or irrigation.

Numerous studies have explored the quality of the harvested water from different roof types and for different types of contaminates, including heavy metals [23,24], organic contaminants [25,26], inorganic substances and suspended solids [27–29], and the presence of microorganisms and fecal coliforms [24,30]. From these works, it appears that different roof types (e.g., asphalt fiberglass shingle, bitumen sheets, metal, concrete tile, green roof) have different impacts on the quality of the harvested water. Nevertheless, most studies have reported low water quality of the first flush in the beginning of the rainy season. It was explained that the first flush washes off the contaminants that were accumulated over the roofs during the dry season. These contaminates may include dust and atmospheric pollutants that settled down, birds and animal feces, and other organic components, such as leaves and dead rodents [26–28,31]. In addition, roof construction materials may also undergo various chemo-physical degradation process, which may lead to the release of various pollutants to the harvested rainwater [32–34].

### 1.3. Bitumen Sheets and Water Quality

Bitumen is a black adhesive material produced from crude oil. Currently, roof sealing with bituminous products is a very common practice worldwide, with hundreds of thousands of tons of bitumen being used for this purpose [35]. Several works have indicated the high potential of roofs covered by different types of petroleum bitumen to deteriorate harvested water quality, including roofs covered with bitumen sheets [24–26,29,36,37]. Bitumen sheets are flexible layers, typically 4–8 mm thick, composed of bitumen mixed with different polymers and reinforcing materials, and their use for roof sealing, in modern buildings is very popular. For example, in Germany, the manufacturing of bitumen products for roofing membranes and waterproofing sheets represents the second-largest

application for bitumen, following road construction [38]. Many bitumen products may emit polycyclic aromatic hydrocarbons that have mutagenic and carcinogenic properties [29,37,38]. Bucheli et al. [25] detected high levels of the R-mecoprop herbicide and its S-enantiomer in runoff water from bitumen roofs. It was hypothesized that the herbicide was added to the bitumen to prevent root penetration and plant growth over the roofs. In addition, bitumen roofs exhibited high emission rates of inorganic components, including major ions (e.g., Ca, K, Mg, Na, P, and S) and metals [29,36–38].

The Bitumen sheets (BS) on top of the roofs are exposed to diverse and sometimes harsh environmental conditions. These conditions include high temperatures during the summer months, high moisture, and low temperatures during the winter. In addition, many roofs are exposed to high levels of sun radiation. Several works have studied the impact of these environmental conditions on the degradation and accelerated aging processes of the BS. However, these works have focused on the physical and mechanical properties of the BS [39–42], ignoring the environmental aspects that these degradation processes may have, such as the emission of pollutants to the environment. Since the main purpose of BS is to physically seal the roof surfaces, it is well understood why the physical properties of the sheets are so important and well-studied. However, as the rooftops start to act as operational surfaces to harvest rainwater, more aspects of the BS need to be explored.

As detailed above, several works have indicated the high potential of BS to emit solutes and toxic substances to the environment. However, the impact of the environmental conditions on the release of these substances and their transport dynamics by harvested rainwater are still far from being fully understood. Recently, Müller et al. [43] discussed the potential impact that environmental conditions, mainly precipitation and temperatures, may have on emission of different substances and pollutants from different roof types, including roofs covered by bituminous products. This study attempts to shed more light on the dynamics and processes of solutes emission from BS to harvested rainwater under different environmental conditions, focusing on the impact of sun radiation on solutes emission.

## 2. Materials and Methods

This study consists of the following parts: (1) water sampling (survey) of the harvested rainwater from BS-covered roofs, and (2) a controlled experimental study to characterize the emission of different solutes from BS to the water phase under various environmental conditions. Herein, the terms 'roof water' and 'leachate' are used to denote water that interacted with BS on rooftops and was surveyed, and water that interacted with the BS in the experimental section of the work, respectively.

### 2.1. Roof Water Survey, Sampling and Analysis

The initial evaluation of the quality of harvested water from roofs covered with BS was conducted by sampling roof water from gutters of four BS-covered roofs in the center of Israel during the winter of 2019–2020 (Table 1). The average annual precipitation in the region is 550 mm, and extreme storm events may have high precipitation rates in the order of 100 mm/d (~once every 10 years). The rainy season usually begins in October and ends in April. Water samples were collected during the very first flush of the season at the first rain event (19 October 2019) from all roofs, excluding roof #3. Throughout the rest of the season, samples were collected in the middle of each rain event, excluding roof #1, where water samples were taken in the beginning and the end of each event. In addition, at the first rain event of the season, water samples were collected from roof #1 every 30–60 min to observe the temporal changes of water quality throughout the entire event. The precipitation depth was measured at the sites of roofs #1 and #3 using a tipping bucket rain gauge (RainWise RainLogger, Rainfall Data Logging System, Boothwyn, PA, USA). In addition, rainwater samples were collected in February 2020, and the electric conductivity (EC) values in the rainwater samples were determined by EC meter. The EC

values indicated the electrolyte concentrations in the water samples, which correlated with the salinity degree of the samples.

**Table 1.** Locations of the four roofs used for water sampling.

| # | Location | Description | Coordinates | Period | Roof Age |
|---|----------|-------------|-------------|--------|----------|
| 1 | Givat-Brener | Private house | 31°52′15.5″ N 34°48′01.1″ E | 19 October 2019–10 February 2020 | 13 years |
| 2 | Rishon-LeTsiyon-1 | Industrial building | 31°59′30.6″ N 34°49′12.2″ E | 19 October 2019–10 January 2020 | >15 years |
| 3 | Rishon-LeTsiyon-2 | Industrial building | 31°59′28.2″ N 34°49′13.4″ E | 30 November 2019–10 January 2020 | >15 years |
| 4 | Nezer-Sereni | Private house | 31°55′26.9″ N 34°49′26.3″ E | 19 October 2019–10 January 2020 | 1 year |

The collected roof water was filtered through a 0.22 μm filter and analyzed for concentrations of Na, Ca, Cl, total organic carbon (TOC), and EC. The total dissolved solids concentration (TDS) was calculated upon EC readings, as detailed by Rhoades [44]. The instruments used for the chemical analyses are detailed in Table 2.

**Table 2.** Examined elements in the water samples and analytical tools.

| Examined Element | Analytical Tool |
|------------------|-----------------|
| Na | Flame photometer (Model 420) Clinical Flame Photometer, Sherwood Scientific Ltd., Cambridge, UK |
| Ca | Absorption Spectrometer Aanalyst 400, PerkinElmer Inc., Waltham, MA, USA |
| DOC/TOC | Total Organic Carbon Analyzer (TOC-L) + Total Nitrogen Measuring Unit (TNM-L), Shimadzu, Kyoto, Japan |
| EC | pH, mv, Conductivity/TDS/C°/F°, Eutech PC 450, Thermo-Fisher, Waltham, MA, USA |
| Turbidity | Turbidimeter, Eutech TN-100, Thermo scientific, Waltham, MA, USA |
| Metals | 720-ES ICP Optical Emission Spectrometer (ICP-OES), Varian, CA, USA |

*2.2. Controlled Experimental Study*

An industrial bitumen sheet, which is used worldwide for roof sealing, was used in this study (BITUMPLAST, 4 mm white, Vyberg, Russia). Pieces of the BS were placed under different environmental conditions for varied time periods and immersed into distilled water (DI) to examine the release of different substances from the BS to the water phase. The four examined environmental conditions included: (i) open air in the laboratory, with no exposure to direct sun light. Herein, this will be referred to as 'Lab'; (ii) in a dark oven at 40 °C. Herein, this will be referred to as 'Oven'; (iii) on the roof of the laboratory (roof #3 in Table 1), shaded by a wooden surface positioned 5 cm above the BS. Herein, this will be referred to as 'Shade'; and (iv) in adjacent to the 'Shade' samples on the roof, with no shading. Herein, this will be referred to as 'Sun'.

The BS pieces were 2 cm wide and 7 cm long. In total, 12 BS pieces were placed in the different environmental setups, as detailed above. On 9 May 2021 and every week, for 4 consecutive weeks, three pieces (three replicas) were taken to the laboratory for analysis, as depicted in Table 3. For the analysis, each one of the BS pieces was inserted into a 50 mL plastic tube, which was filled with DI water, and then shaken for 3 h. Following this, the

leachate water from each tube was filtered through a 0.22 μm filter and the EC, turbidity, and dissolved organic carbon (DOC) concentration were measured, as detailed in Table 2.

**Table 3.** Time scheme of the BS experiments. Non-bold and bold checkmarks (√//√) indicate operations on the small and large BS pieces, respectively.

| | | 9 May 2021 | 16 May 2021 | 23 May 2021 | 30 May 2021 | 6 June 2021 | 6 July 2021 | 9 September 2021 |
|---|---|---|---|---|---|---|---|---|
| **Temporal Information** | **Date** | 9 May 2021 | 16 May 2021 | 23 May 2021 | 30 May 2021 | 6 June 2021 | 6 July 2021 | 9 September 2021 |
| | **days from T0** | 0 | 7 | 14 | 21 | 28 | 58 | 123 |
| | **Days from last exposure to rain** | - - - - - - - - - | - - - - - - - - - | - - - - - - - - - | - - - - - - - - - | - - - - - - - - - | 30 | 65 |
| **Action** | **Small pieces 7 cm × 2 cm** | 12 pieces (for each setup) put in place | Three pieces from each setup taken for analysis | Three pieces from each setup taken for analysis | Three pieces from each setup taken for analysis | Three pieces from each setup taken for analysis | - - - - - - - - - | - - - - - - - - - |
| | **Large pieces 30 cm × 50 cm** | All pieces put in place | - - - - - - - - - | - - - - - - - - - | - - - - - - - - - | All pieces taken to rain simulator and returned to place | All pieces taken to rain simulator and returned to place | All pieces taken to rain simulator |
| **Analyses** | **EC** | √ | √ | √ | √ | √//√ | √ | √ |
| | **Turbidity** | √ | √ | √ | √ | √//√ | √ | √ |
| | **DOC** | √ | √ | √ | √ | √//√ | - - - - - - - - - | - - - - - - - - - |
| | **Metals** | - - - - - - - - - | - - - - - - - - - | - - - - - - - - - | - - - - - - - - - | √ | - - - - - - - - - | - - - - - - - - - |
| | **Major ions** | √ | √ | √ | √ | √//√ | √ | √ |

In addition, larger BS, 50 cm long and 30 cm wide each, were positioned next to the small pieces for a longer period of time and used to study the release of the substances under conditions of a simulated rain event (details below). These big BS pieces were tested over three repeated rain events, with dry periods of several weeks in between. Following each simulated rain event, the BS pieces were returned to their locations at the different environmental setups. Table 3 details the time scheme of the large BS experiment. The simulated rain events were conducted in a Morin-type rainfall simulator, which enables the simulation of rainstorms with different rainfall intensities on inclined surfaces [45]. The simulated rain events were of a 43.0 mm cumulative rain, with rain intensity of 48.0 mm/h. During the simulations, the BS were positioned in slopes of 5°, and the surface runoff water was sampled every few seconds during the first ~10 min of the simulation, as well as every few minutes following that. The collected leachate samples were filtered and analyzed for EC, DOC, and turbidity, as detailed above and in Table 2.

As aforementioned, sun radiation is assumed to have an important role in the degradation processes of the BS [40,41,46], which may affect the release of different components to the roof water. Therefore, the 'Sun' and 'Shade' setups were examined under the conditions of exposure to direct and indirect sun radiation, which were measured throughout the period of the study at a meteorological station of the Israeli Meteorological Service, located ~1 km north of roof #3, where the experiment was conducted (data available on line https://ims.data.gov.il/he/ims/6 (accessed on 13 September 2021)).

### 3. Results and Discussion

*3.1. Roof Water Survey*

The sample water from the different roofs indicated a clear pattern of high load of solutes at the first flush of the first rain event of the season, followed by a sharp reduction of salinity and concentrations of the different elements in the following rain events of the season (Figure 1). The EC readings of the first flush samples were two orders of magnitude higher than sampled rainwater EC (~0.04 dS/m) and the EC reading of the following sampled roof water during the winter (Figure 1F). The same trend was observed for all

other examined elements (Figure 1). It was also visually observed that the first rain event generated brownish runoff water from the BS-covered roofs (Figure 2), which was not observed during the following rain events. This color is likely associated with the release of organic components from the BS, which correspond to the high measures of TOC at the first rain event (Figure 1E).

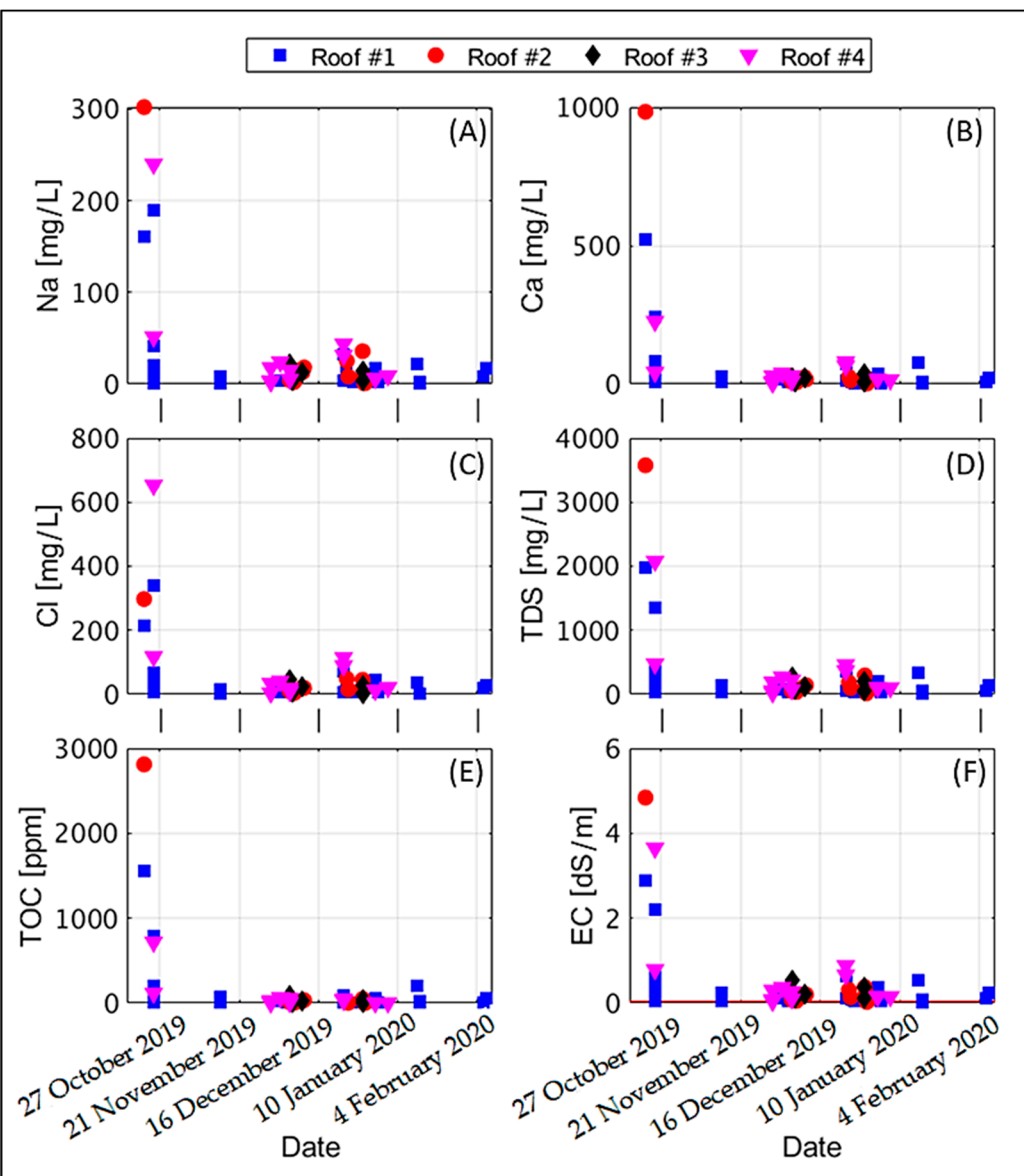

**Figure 1.** Measured concentrations of sodium (**A**), calcium (**B**), chlorine (**C**), TDS (**D**), TOC (**E**), and the electric conductivity (**F**) of the sampled water from the examined four roofs. The red line in (**F**) marks the measured EC of the rainwater sample (~0.04 dS/m). Each data point is of a single sample.

The high loads of dissolved organic components in the first flush of the rainy season reflect the buildup of soluble components on the rooftops during the long dry summer, which were dissolved and washed off the roof by the first flush during the first rain event. As detailed above, roof #1 was sampled in high temporal resolution so that the temporal changes of water quality throughout the first rain event could be observed. Figure 3 presents the reduction of sampled water EC during the first rain event of 19 October 2019. The event length was in the order of 6 h, and the cumulative precipitation was equal to 17.0 mm. This collection of EC measurements indicates the very high solute concentration of the first flush and the relatively rapid reduction of solute concentration with time. After

~40 min of continuous rain, the EC readings were reduced by more than 60%, and after ~5 h, the measured EC was equal to the measured rainwater EC. In Figure 3, pictures of the water sampling bottles are also presented. In parallel to the reduction of the sampled water salinity, the turbidity of the water was also reduced. The first sampled water had a strong brown color and, with time, the water became cleaner and more transparent.

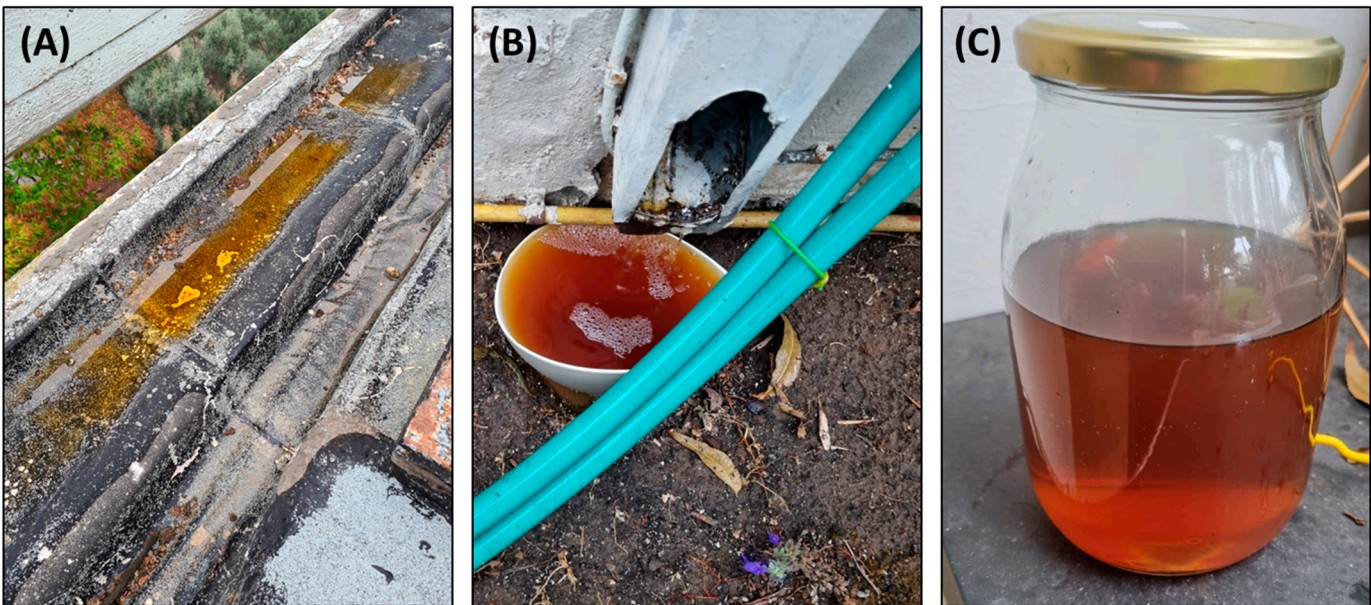

**Figure 2.** Samples of the brownish roof water obtained at the very first flush. (**A**) Ponded water at the rooftop, (**B**) drained brownish water in the gutter, and (**C**) collected water at the first flush of the season.

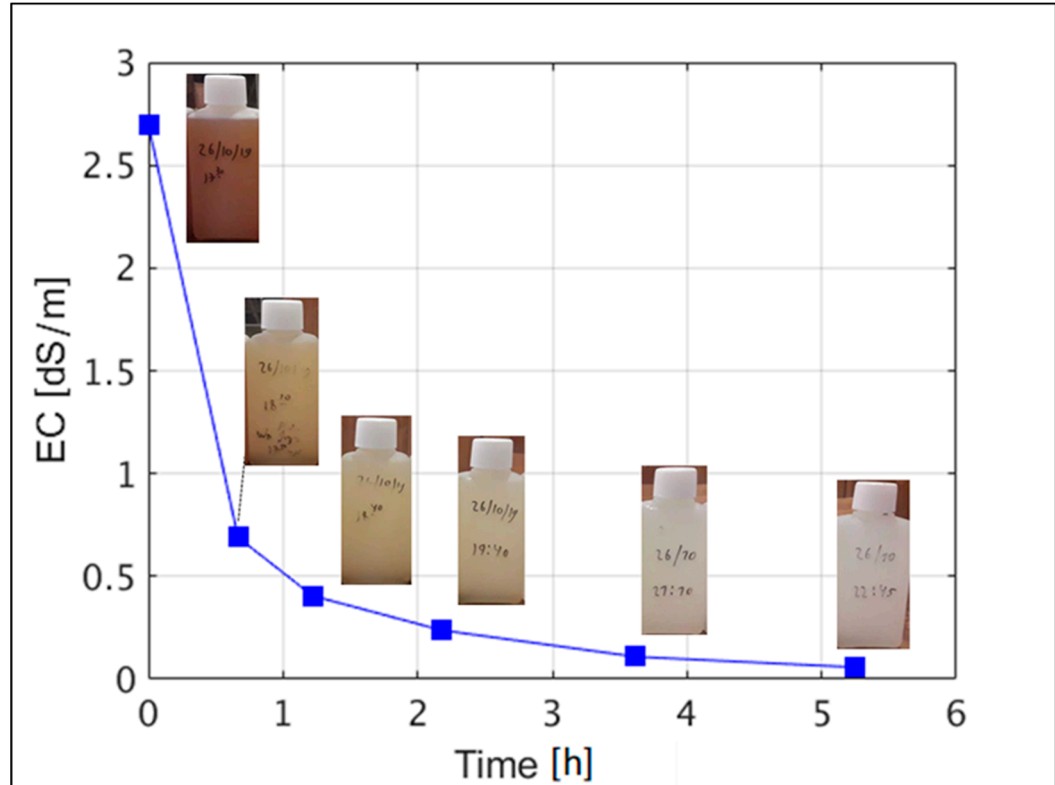

**Figure 3.** Measured EC in the water collected from roof #1 at the first flush of the season. Time is from onset of the rain event. The pictured bottles are of the collected water throughout the event to demonstrate the changes in water turbidity with time.

Monitoring the runoff water from roof #1, over a timespan of 120 days, from October 2019 to January 2020, enabled us to observe that the first flush of each rain event was slightly more saline than the last flush of the previous event (Figure 4). Similar to the very first flush of the season, but in a lower magnitude, the short dry periods between the rain events had a similar effect of buildup of soluble components on the rooftops, which were dissolved and transported with the drained roof water during rain events.

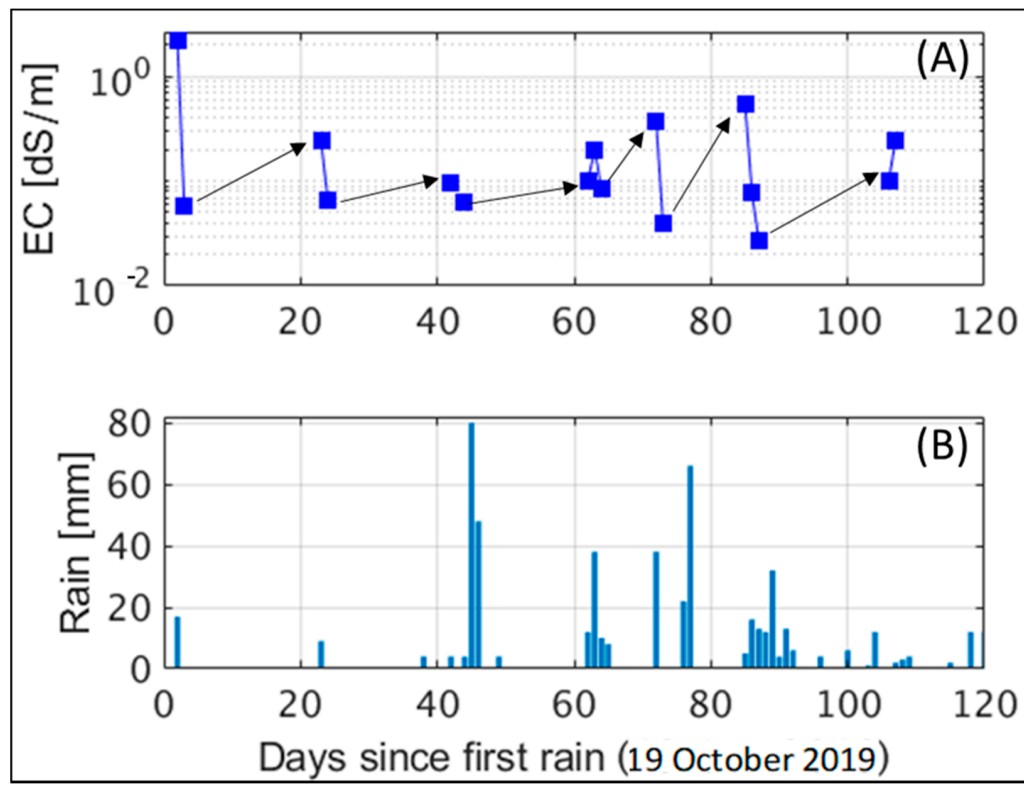

**Figure 4.** Sampled water EC from roof #1 (**A**), and associated rain depth data (**B**).

The components that accumulate on the rooftops, as detailed above, may be a result of dust and other atmospheric pollutant sedimentation, bird and animal feces, and the accumulation of other organic components, such as leaves, dead rodents, and birds. In addition, the degradation processes of the BS used to seal the roofs may also lead to the release of organic and inorganic solutes from the BS to the liquid water phase.

Whereas the accumulation of dust and atmospheric pollutants, bird and animal feces, and other organic components is relatively well understood and documented in the literature [26–28,31], our understanding of the BS degradation processes is more limited. Therefore, the following experiments were conducted to shed more light on the environmental conditions that accelerate emission of different substances from the BS to the roof water and to understand the dynamics of removal of these solutes from the roof by the drained roof water.

### 3.2. Experimental Study

As detailed above, the experimental study included a set of controlled experiments with the small BS pieces, which aimed to test the impact of various environmental conditions on the release of different substances from the BS to leachate water. In addition, the large BS pieces were used with aim to better understand the dynamics of the 'first flush', and the release and removal of the different solutes from the BS by the flowing rainwater.

### 3.2.1. Small BS Pieces

As detailed above, the BS were exposed to conditions of direct sunlight on the rooftop ('Sun'), indirect sunlight on the rooftop ('Shade'), open air in the laboratory ('Lab'), and 40 °C in an oven ('Oven'). The 'Sun' and 'Shade' BS were exposed to direct and indirect radiation conditions and air temperatures as presented in Figure 5.

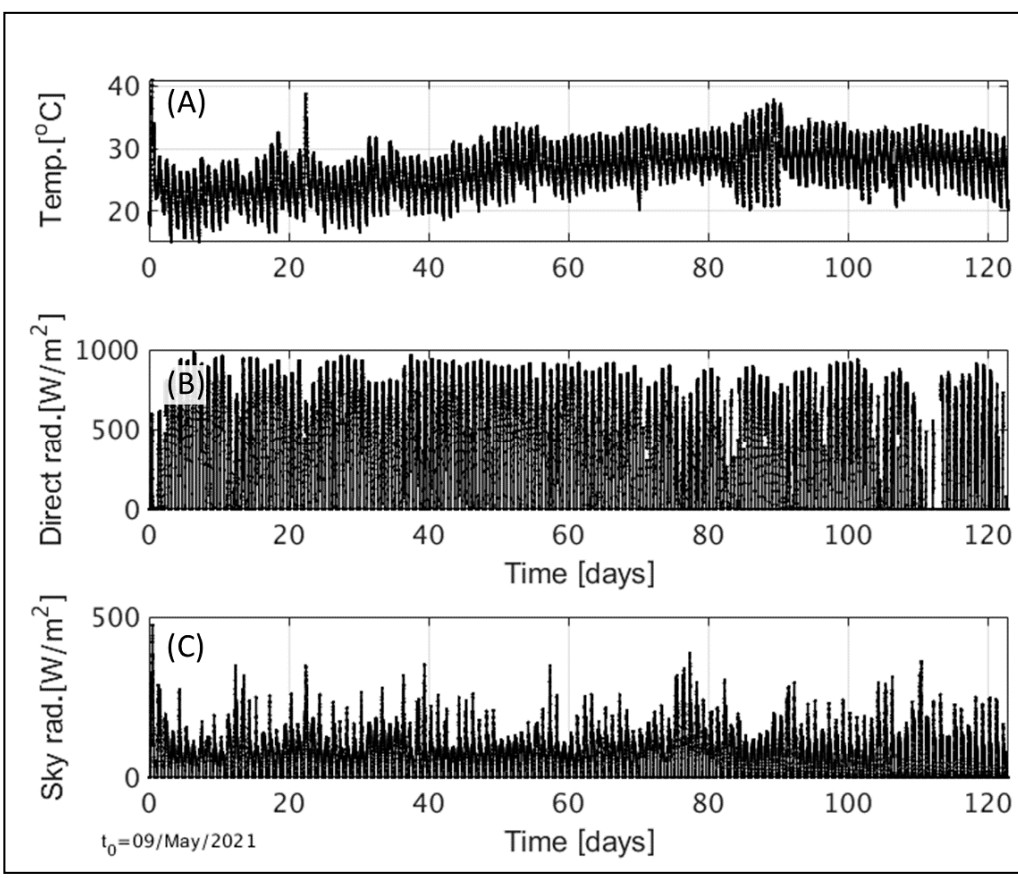

**Figure 5.** Environmental conditions during the experiment, measured ~1 km north of the roof location, from 9 May 2021 to 9 September 2021. (**A**) Temperature in shade, (**B**) direct sun radiation, and (**C**) diffuse sky radiation.

The measurements of leachate water EC demonstrated the high impact that the different environmental conditions had on the release of solutes from the BS to the liquid water phase. The highest EC levels were observed in the 'Sun' BS, followed by the 'Shade' setup (Figure 6). The 'Lab' and 'Oven' samples had the lowest levels of solutes released to the water (Figure 6). In addition, time also affected solute release to the leachate water, as a constant increase in solute emission from the BS to the water was observed for all treatments, most prominently for the 'Sun' treatment. The EC readings of the pieces exposed to the sun for 4 weeks were almost three-fold higher than the initial conditions, whereas for the 'Shade' setup, the increase over 4 weeks was in the order of 1.6 folds. For the 'Lab' and 'Oven' setups, it was even lower, in the order of 1.3 folds (Figure 6).

These results indicate that the BS exposed to direct sun released the highest loads of solutes to the leachate water. It is assumed that this is a result of the accelerated chemical and physical degradation processes that occur at the BS surface due to exposure to UV radiation that readily breaks the chemical bonds within the bitumen polymer chains [46]. The higher measured EC values of the 'Shade' compared to the 'Lab' and 'Oven' setups were likely a result of the 'Shade' exposure to indirect radiation. However, both 'Sun' and 'Shade' were exposed to the open atmosphere. Therefore, dust and other atmospheric pollutants could have settled on these pieces and increased the load of solutes at the

leachate water. To strengthen the notion that the increased EC of these BS was a result of the bitumen degradation processes and not a result of dust accumulation, changes in the DOC and metal emissions were also examined. The sources of these components are less likely to be associated to dust sedimentation, and it is well known from the literature that BS may emit metals and organic components [23–26,47], as well as other inorganic components, as a result of the aging and degradation processes [48].

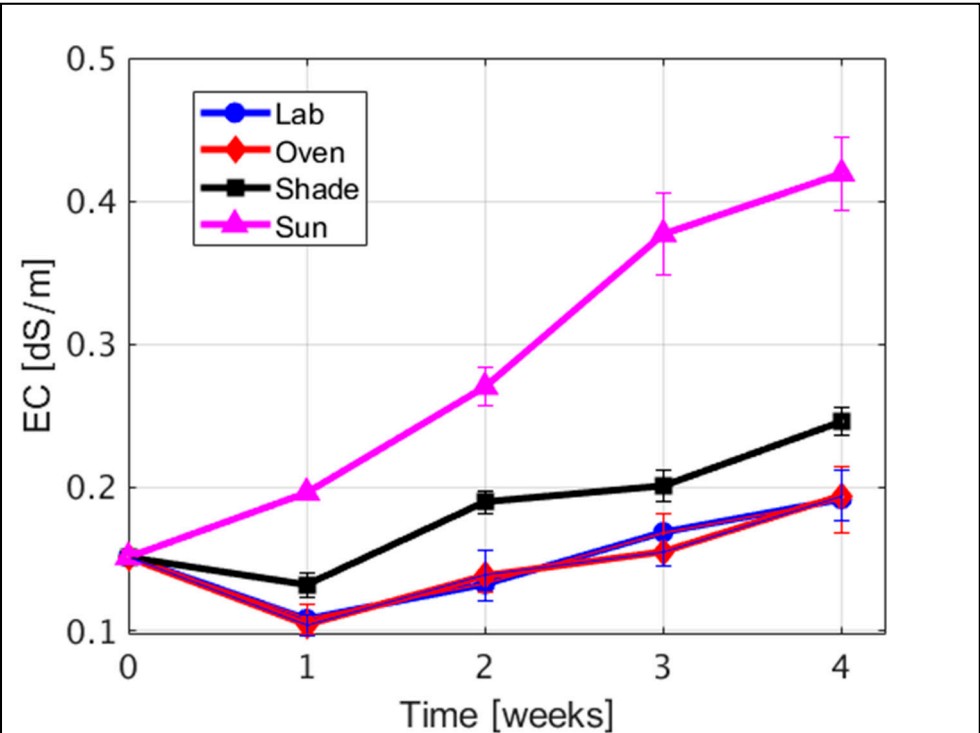

**Figure 6.** Temporal changes in the EC readings of the leachate water from the 'Lab', 'Oven', 'Shade', and 'Sun' bitumen sheets.

The DOC levels in the leachate water were examined every week for the 'Sun' setup and at the end of the fourth week for all setups. It is seen that after 4 weeks, the 'Sun' setup DOC levels were greater than 600 ppm, followed by the 'Shade' setup, which was in an order of magnitude lower at the order of 60 ppm. The 'Lab' and 'Oven' setups had the lowest DOC reading in the order of 3 and 1.5 ppm, respectively (Figure 7). Similar to the sampled roof water, the 'Sun' leachate water had a yellowish color, which is associated with high organic loads and elevated DOC levels [23]. In concordance with the EC readings, the DOC levels of the sun-exposed BS pieces increased gradually as the exposure of the BS to the sun was longer. A linear correlation ($R^2 = 0.94$) was found between the EC and DOC concentrations of the 'Sun' leachate water (Figure 8), strengthening the assumption that increased EC readings of the 'Sun' BS are associated with chemo-physical degradation processes, and not with the dust accumulation on the examined sheets.

The measured metals concentrations in the leachate water of all examined environmental setups also indicated the negative effect that direct sun radiation had on the degradation processes of the BS, which resulted in the release of metals to the water phase (Figure 9). The emission of Boron (B) was relatively equal for all setups, with observed B levels in the order of 10–20 ppb in the leachate water. Copper (Cu), iron (Fe), and zinc (Zn) were not emitted from the 'Lab' and 'Oven' setups, while for the 'Shade' setup, the measured concentrations of these metals were in the range of 2–10 ppb (Figure 9). For the 'Sun' setup, the Cu, Fe, and Zn concentrations were equal to ~10, 100, and 20 ppb, respectively. Manganese (Mn) was observed to be emitted only from the 'Sun' setup with concentrations in the order of 10 ppb. Even though the overall metal concentrations were relatively low, it

is clearly seen that direct sun radiation, and to some extent the indirect sun radiation, led to the elevated emission of metals from the BS to the water phase.

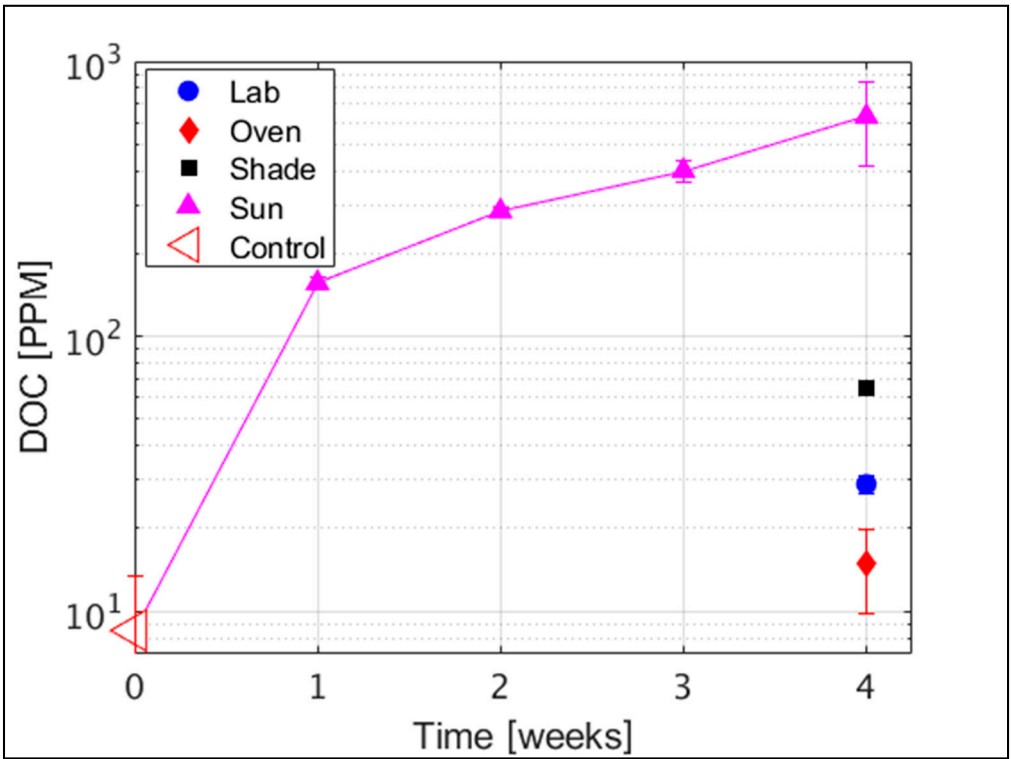

**Figure 7.** DOC concentrations for the 'Sun' setup throughout the entire length of the experiment and for the 'Lab', 'Oven', and 'Shade' setups at the fourth week of the experiment.

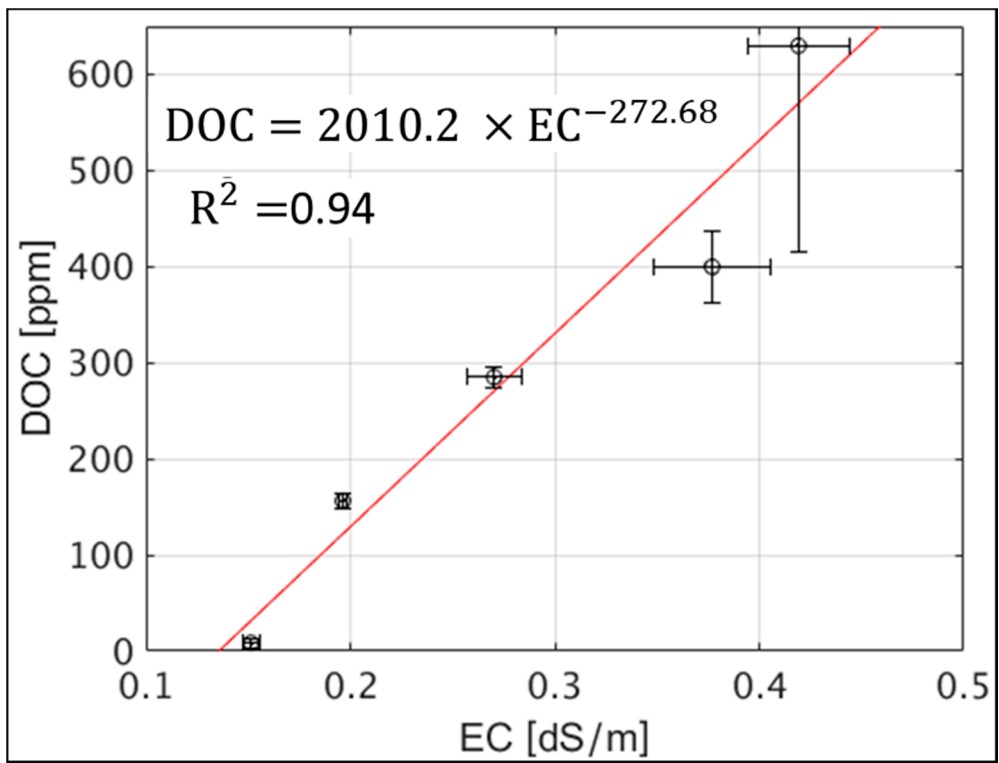

**Figure 8.** Linear correlation between the EC and DOC readings of the 'Sun' leachate water.

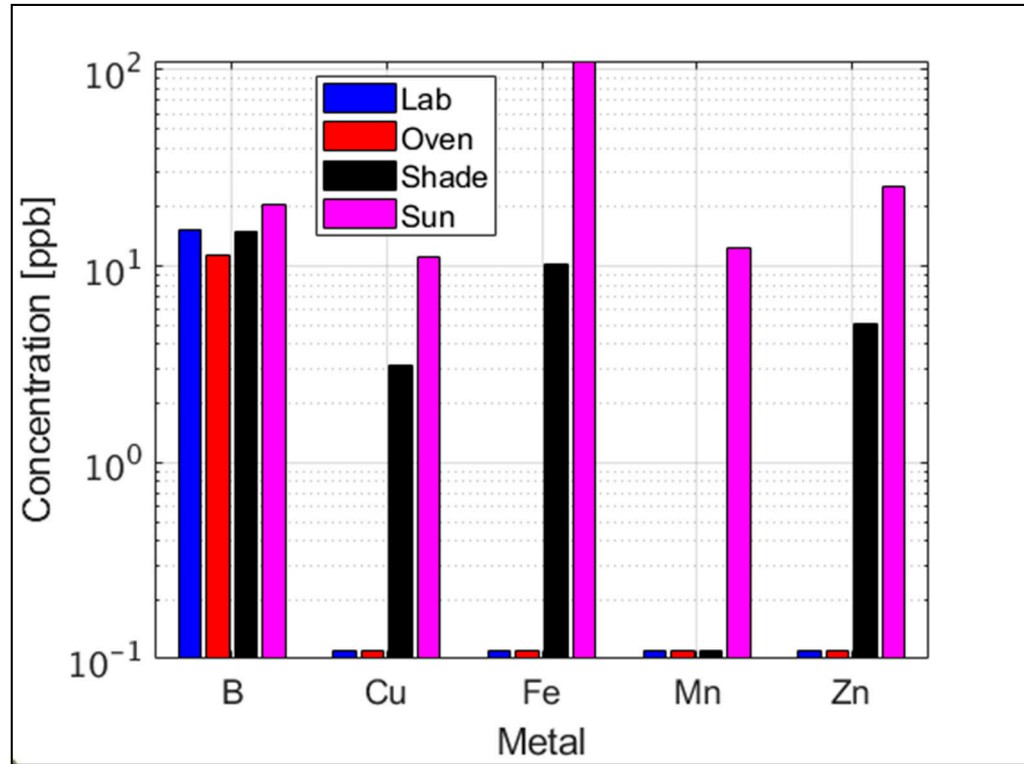

**Figure 9.** Metals concentrations at leachate water from all examined setups in the fourth week of the experiment. Values of $10^{-1}$ ppb means readings were below the detection limit.

The set of observations detailed above demonstrates the strong effect that direct sunlight radiation had on the degradation processes of the BS and the resulting emissions of various organic and inorganic components. However, the observations did not contribute any information about the dynamics of the release of these substances to rainwater that flows over the BS, and the removal of these substances from the roof with the drained water. For this purpose, simulated rain experiments with the larger BS pieces were conducted.

### 3.2.2. Rain Simulation Experiments

The measured EC values of the drained water from the large BS pieces that were exposed to simulated rain are in agreement with both the small BS pieces experiments and the field measurements of the different roofs. The highest EC (1.25 dS/m) was measured in the very first flush of the first simulated rain event, with a cumulative rain of 1.5 mm for the 'Sun' setup (Figure 10A). This was followed by the 'Shade' setup, where the measured EC was equal to 0.67 dS/m. The 'Lab' and 'Oven' readings were the lowest, with EC readings of 0.48 and 0.38 dS/m, respectively. For all setups, a notable reduction in the EC was observed during the rain event, with the stabilization of drained water EC values, as well as turbidity (Figure 10D), after a cumulative rainfall of ~20 mm. This is in good agreement with the detailed measurements of roof #1, which showed that the roof water was solute-free and clear after 4–5 h of precipitation, with a cumulative rain depth of approximately 20 mm (Figure 4). The first flush salinity of roof #1 was ~50% higher than the first flush of the 'Sun' BS in the first simulated rain event. This is likely a result of the longer exposure of the roof #1 bitumen to the sun throughout the whole antecedent Mediterranean summer, during which both degradation processes of the bitumen occurred, together with the sedimentation of dust and other substances.

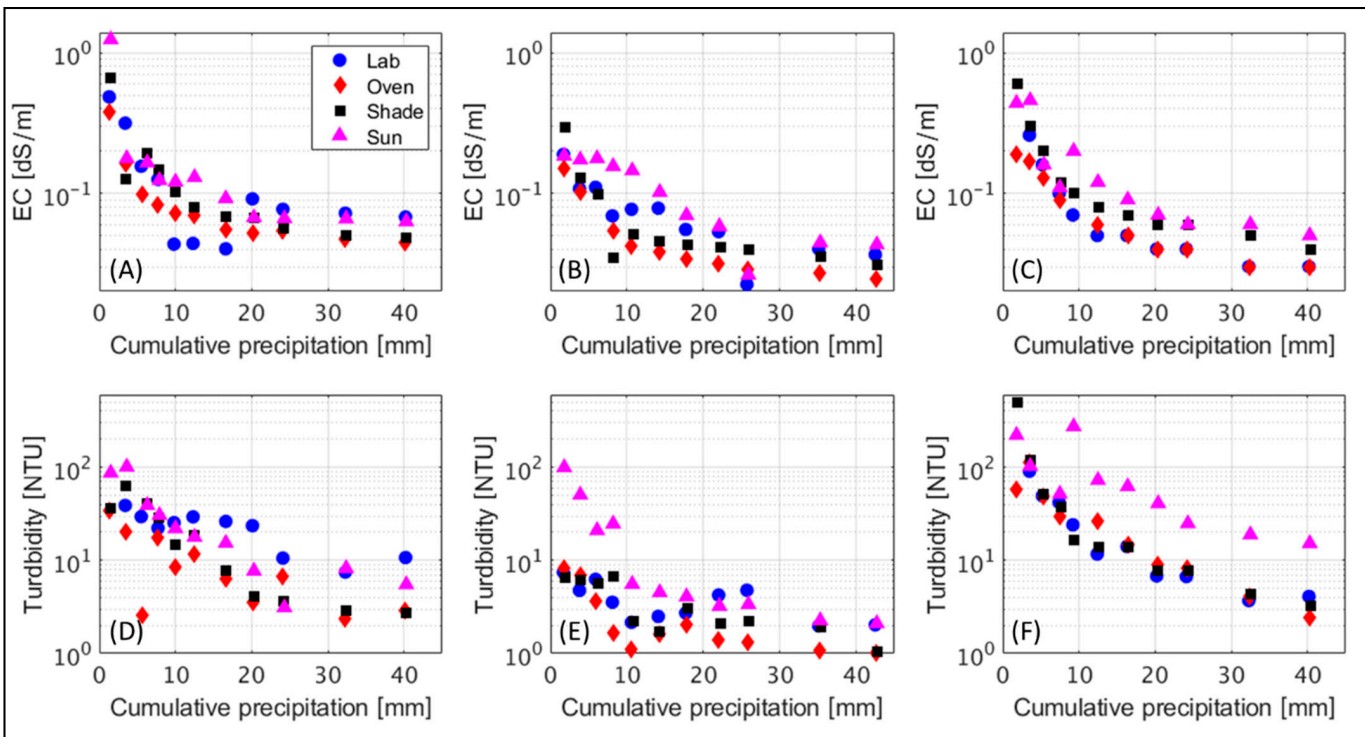

**Figure 10.** Measured EC (**A**–**C**) and turbidity (**D**–**F**) of drained water obtained for all examined setups at the first rain simulation (**A**,**D**), second rain simulation (**B**,**E**), and third rain simulation (**C**,**F**).

At the first flush of the second simulated rain event, which occurred 30 days after the first event (Table 3), a cumulative precipitation of 2 mm generated EC levels of 0.3, 0.2, 0.2, and 0.15 for the 'Shade', 'Sun', 'Lab', and 'Oven' setups, respectively (Figure 10B). Even though first flush salinity of the 'Shade' setup was higher than the 'Sun' setup, the reduction of the 'Shade' salinity with the proceeding of the rain event was faster than the 'Sun' setup, which maintained relatively high and constant EC levels up to a cumulative rainfall of ~10 mm. All other setups, including 'Shade', presented a relatively rapid reduction in salinity of the drained water, with a notable reduction measured after a cumulative rainfall of 4 mm. The 'Oven' and 'Shade' setups reached and stabilized at the lowered salinity levels, of ~0.05 dS/m, after about 10 mm of rain, whereas the 'Lab' and 'Sun' reached these EC levels after a cumulative rainfall of 17 and 22 mm, respectively. In respect to drainage turbidity, the 'Sun' setup drainage was an order of magnitude higher than all other setups up to a cumulative rainfall of 4 mm. It was only after 10 mm of cumulative rain when the turbidity of the 'Sun' setup decreased to the turbidity levels of other setups.

A third simulated rain event occurred after an additional 65 days of dry conditions under the different environmental conditions. The EC levels of all treatments were higher than their counterparts in the second simulated rain event but lower compared to the first event. This is likely a result of the longer dry period between the second and third simulated rain events compared with the 30-day dry period between the first and second rain events.

The DOC levels were measured in the drained water from all BS setups during the first simulated rain event at the first flush, after about 2 mm of cumulative rain where salinity and turbidity were maximal, and after a cumulative rainfall of ~30 mm, when salinity and turbidity were low and stable. Figure 11 presents the measured DOC levels, and it is clearly seen that the sun-exposed BS released the highest levels of DOC to the drained water. In agreement with the above-mentioned observations, this again indicates the major effect that sun radiation has on chemo-physical processes at the BS surface, which led to enhanced aging and degradation processes of the SB. In turn, this led to the elevated release of organic and inorganic solutes to the liquid water that flows over the BS.

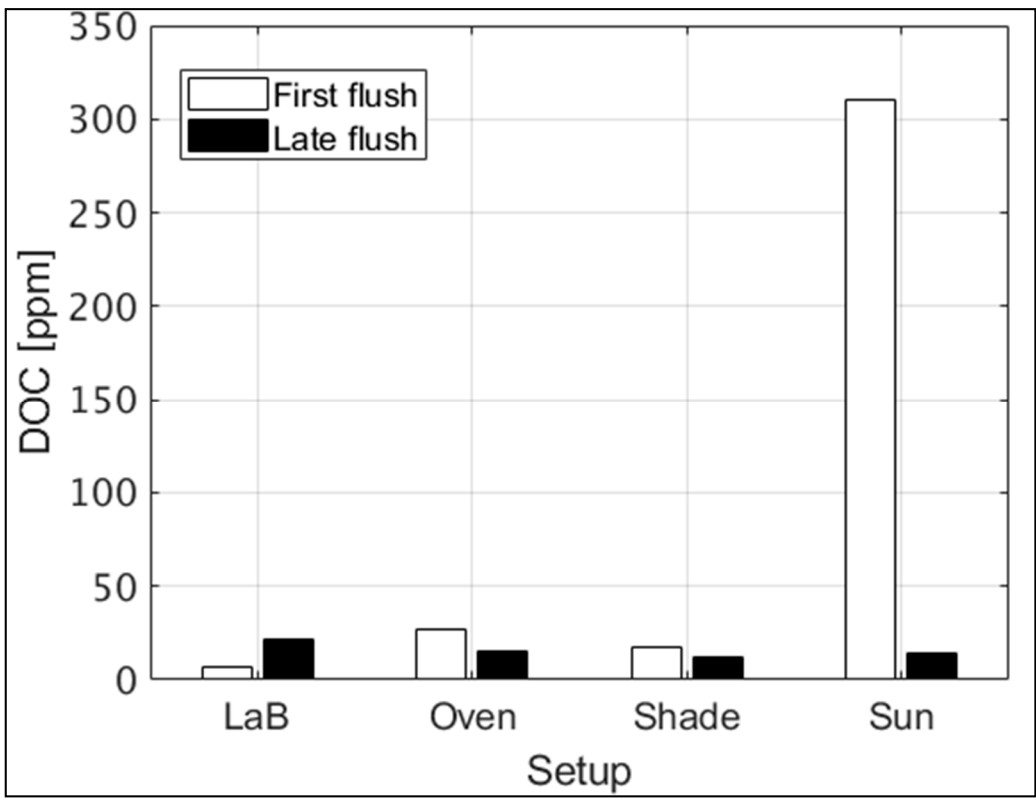

**Figure 11.** DOC concentrations in drained water from the different BS setups at the beginning of the rain event (first flush) and toward the end.

### *3.3. Results Integration*

Both the field survey and experiments indicated the high loads of solutes that were leached from the BS at the first flush of the rainy season (Figures 1, 3, 4 and 10). It was shown that sun-exposed sheets emitted the highest loads of organic and inorganic solutes and metals compared to all other examined environmental conditions (Figures 6, 7 and 9–11), likely a result of the accelerated aging and degradation processes of the sun-exposed bitumen sheets [46]. As detailed above, BS may emit toxic substances, such as polycyclic aromatic hydrocarbons and volatile organic compounds [24], which were not examined in this work. It should be validated by future studies, but it is believed that the release dynamics of these toxic substances would be in agreement with the findings of this work, i.e., high concentrations at the first flush, negligible emission of the toxic components at the following rain events, and the highest emission from BS exposed to direct sunlight.

As aforementioned, the high loads of solutes in the first flush were a result of the accumulation of soluble components during the summer on the rooftops, a result of degradation processes of the BS, and the sedimentation of dust and other pollutants. More studies are needed to better characterize the runoff properties from different roofs as a function of roof cover, slope, and other environmental conditions. However, for the conditions examined here, of a flat or low-slope roof covered with BS, both the field measurements and rain simulator experiments showed that a cumulative rainfall of about 20 mm was sufficient to wash off the solutes that accumulated over the roof during the summer (Figures 1 and 10). The rain simulator experiments conducted herein (Figure 10) showed that a dry period of about 30 days is sufficient to result in a notable release of solutes from the bitumen to the roof water, mainly for the conditions of roofs that are exposed to direct sunlight. However, good quality roof water was measured for shorter dry periods between the rain events, at the order of 20 days, as observed for the surveyed roofs (Figures 1 and 4).

The sedimentation of the different pollutants over the rooftops and the aging and degradation processes of the BS are processes that occur at the upper surface of the BS. These processes arise every long dry period, regardless of the roof washing processes that may have occurred during the preceding winters and rain events. Therefore, a reduction of pollutants emission from BS-covered roofs over the years is not expected. This assumption is supported by the high loads of solutes that were measured at the four roofs monitored in the survey, which are all older than 10 years, excluding roof #4 (Table 1). Future studies should clarify this point, but old BS may emit higher loads of solutes and pollutants due to the natural aging of the BS, which accelerates the chemo-physical degradation process that leads to the release of various pollutants to the harvested rainwater [32–34]. This may be another explanation for the higher EC values measured at the old roofs that were surveyed compared to the EC readings of the first flushes generated in the rain simulator experiments from a new BS (Figures 1F and 10A–C).

## 4. Summary and Conclusions

This work aimed to shed more light on the processes of emission and transport of organic and inorganic solutes from bitumen sheets used for roof sealing. Studying these processes is important, as rainwater harvesting from rooftops in urban environments is becoming a common practice in many cities worldwide, and the characterization of the harvested water quality has great importance. Even though bitumen sheets are widely used to seal rooftops in modern cities, our understanding of the impact of these surfaces on the quality of harvested water from the roofs is still limited.

In this study, four bitumen-covered roofs in the center of Israel were monitored over the winter of 2019–2020 to characterize concentrations of organic and inorganic solutes in the roof water throughout a series of consecutive rain events. Following this, two series of controlled experiments were conducted to better understand the impact of different environmental conditions on the emission of solutes and metals from the bitumen sheets to the water phase that flows over them during rain events. The examined environmental conditions were of: (i) open air in the laboratory with no direct sun radiation; (ii) 40 °C in a dark oven; (iii) shaded conditions on a rooftop; and (iv) sun exposed bitumen sheets on the same roof.

Both the field survey and experiments indicated the high loads of solutes being flushed off the bitumen sheets at the first flush of the rainy season. It was shown that sun-exposed bitumen sheets emitted the highest loads of organic and inorganic solutes and metals compared to all other examined environmental conditions.

These research findings may suggest that for water quality consideration, rooftops that are covered with bitumen sheets and used for rainwater harvesting should be shaded by different means to reduce the release of the different pollutants. Shading methods may include the spreading of a porous medium such as gravel at a thickness of a few centimeters on top of the bitumen sheets. Other options may be the use of more environmental and comprehensive solutions, such as green roofs, or shading of the bitumen sheets by solar panels. In addition, engineers and designers of rainwater harvesting systems may consider different ways to divert the first flush water to the municipal drainage system and not to harvest this water, which may be highly contaminated. Another solution could be to replace bituminous products by other materials that do not emit pollutants to the environment, or to improve the formulation of the bitumen sheets to be more environmentally friendly.

**Author Contributions:** Writing—original draft, U.N. and L.N.; Writing—review & editing, M.B.-H., D.K., R.K., G.G. and Y.L. All authors have read and agreed to the published version of the manuscript.

**Funding:** This research was funded by The Israel Water Authority, grant #: 4501847430.

**Institutional Review Board Statement:** Not applicable.

**Informed Consent Statement:** Not applicable.

**Data Availability Statement:** Raw data + matlab codes used for analyses can be found at: Nachshon, Uri (2021): BITUMEN DATA. figshare. Dataset. https://doi.org/10.6084/m9.figshare.17134787.v1, (accessed on 17 November 2021).

**Conflicts of Interest:** The authors declare no conflict of interest.

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
