# Peer review of "Dynamic Release of Solutes from Roof Bitumen Sheets Used for Rainwater Harvesting"

_water, doi:10.3390/w13243496_

Round 1
Reviewer 1 Report
The authors addressed the topic of harvesting runoff from roofs covered by bitumen sheets. The study was well designed and executed, and produced the results of interest to the readers of the Water journal. The reviewer would like to draw the authors’ attention to some issues that would enhance their paper discussion, and to a reference addressing similar issues as the authors, but from a different environmental angle (Muller et al., 2019).
1. The authors emphasized the importance of the first flush during the first rain event after an extended dry period, and of high loads of solutes occurring under those circumstances. When this event occurred, was the rainwater interacting with a fresh bitumen material, or was the material “aged” (washed off) prior to this event. In other words, would one expect the same loads of solutes in future years?
2. Some brief description of the rain regime in test location would be helpful when contemplating the use of the data produce elsewhere.
3. Is rainwater pH a potentially influential experimental factor?
4. A minor typo: Line 379: ….Studying these…, rather than ….Studying of these….
5. Reference of potential interest: Muller, A., Osterlund, H., Nordqvist, K., Marsalek, J., Viklander, M. (2019). Building surface materials as sources of micropollutants in building runoff: A pilot study. Science of the Total Environment, 680: 190-197.
Author Response
Reviewer #1
The authors addressed the topic of harvesting runoff from roofs covered by bitumen sheets. The study was well designed and executed, and produced the results of interest to the readers of the Water journal. The reviewer would like to draw the authors’ attention to some issues that would enhance their paper discussion, and to a reference addressing similar issues as the authors, but from a different environmental angle (Muller et al., 2019).
We thank the reviewer for the positive review and constructive comments. Please see our reply to each comment below.
- The authors emphasized the importance of the first flush during the first rain event after an extended dry period, and of high loads of solutes occurring under those circumstances. When this event occurred, was the rainwater interacting with a fresh bitumen material, or was the material “aged” (washed off) prior to this event. In other words, would one expect the same loads of solutes in future years?
At the small BS experiments, the pieces were "fresh", as defined by the reviewer. At the two other parts of the study: (1) Roofs survey; and (2) large BS experiments, the pieces were exposed to consecutive rain events. In other words, results indicate that the release of the different substances is not reduced with time and that long dry periods generate the accumulation and release of high levels of solutes from the BS, mainly after long exposure to sun radiation.
In the revised MS we added data about the roofs ages (Table 1) and shortly discussed this issue [P15, L421-434]
- Some brief description of the rain regime in test location would be helpful when contemplating the use of the data produce elsewhere.
Added at the M&M section [P3, L134-136].
- Is rainwater pH a potentially influential experimental factor?
We measured pH and it was neutral for most cases, and no significant connection was found between pH and other readings. Therefore, for simplicity, we didn't discuss it.
- A minor typo: Line 379: ….Studying these…, rather than ….Studying of these….
Fixed. Thank you.
- Reference of potential interest: Muller, A., Osterlund, H., Nordqvist, K., Marsalek, J., Viklander, M. (2019). Building surface materials as sources of micropollutants in building runoff: A pilot study. Science of the Total Environment, 680: 190-197.
An interesting and relevant paper. We mentioned it at the results and discussion section [P3, L117-120].
Reviewer 2 Report
Dear Authors,
the work deals with an interesting topic and seems to be very current. Research on rainwater management in urban area is obviously necessary, especially in semi-dry and dry regions. Sustainable exploitation of natural resources and their protection is of key importance for smart development. To achieve this, it is necessary to implement alternative sources of water, for example rainwater, in all areas of economy. Rainwater harvesting from roof tops in urban environments is becoming a common practice in many cities worldwide, and characterization of quality of harvested water has great importance. This study presents the results of series of experiments of quality harvested rainwater from buildings roofs located in the center of Israel. In this study, four bitumen covered roofs in the center of Israel were monitored over the winter of 2019-2020 in order to characterize concentrations of organic and inorganic solutes in the roof water. In my opinion, such research is not novelty but the outcomes can be beneficial for engineers and designers of rainwater harvesting systems.
The paper is well-organized, containing all of the expected components. The methodology is effective in attaining the object of this work, but I have question: Why is the radiation data presented for the summer period? The authors performed tests of rainwater quality for winter. Radiation may affect the results.
In the Introduction, the authors provided a brief research background. Summary and Conclusions are supported by the results presented in this paper.
The text requires an editorial correction, for example:
- Description of Table 2
- 2 Experimental study
- Description of all figures - without :
- Please, format the article, especially all tables and references according to the journal's guidelines
Author Response
Reviewer #2
the work deals with an interesting topic and seems to be very current. Research on rainwater management in urban area is obviously necessary, especially in semi-dry and dry regions. Sustainable exploitation of natural resources and their protection is of key importance for smart development. To achieve this, it is necessary to implement alternative sources of water, for example rainwater, in all areas of economy. Rainwater harvesting from roof tops in urban environments is becoming a common practice in many cities worldwide, and characterization of quality of harvested water has great importance. This study presents the results of series of experiments of quality harvested rainwater from buildings roofs located in the center of Israel. In this study, four bitumen covered roofs in the center of Israel were monitored over the winter of 2019-2020 in order to characterize concentrations of organic and inorganic solutes in the roof water. In my opinion, such research is not novelty but the outcomes can be beneficial for engineers and designers of rainwater harvesting systems.
The paper is well-organized, containing all of the expected components. The methodology is effective in attaining the object of this work, but I have question: Why is the radiation data presented for the summer period? The authors performed tests of rainwater quality for winter. Radiation may affect the results.
We thank the reviewer for the positive review.
As for the question about the sun/winter radiation: The fundamental concept of this work (and others) is that the first flush water quality is affected by accumulation of solutes at the roof top during the summer months (the dry period). Therefore, the summer radiation is the important factor that affect the BS degradation processes and resultant release of solutes.
In the Introduction, the authors provided a brief research background. Summary and Conclusions are supported by the results presented in this paper.
The text requires an editorial correction, for example:
- Description of Table 2
DONE
- 2 Experimental study
DONE
- Description of all figures - without :
Done in all figures ":" changed to "."
- Please, format the article, especially all tables and references according to the journal's guidelines
The current version of the paper was reedited and prepared by MDPI editors.
Reviewer 3 Report
The manuscript presents a report on the change of solutes from roof bitumen sheets used for rainwater harvesting. The topic is interesting and brings some information to the readers. However, the manuscript still need comprehensive revision to improve its scientific quality, rather than a report.
- Title needs to be changed, delete “Inorganic and Organic”, because it means all compounds.
- Abstract should be rewritten. The authors need to provide the general results they obtained with direct values, for example, the concentration of the pollutants.
- Introduction should be rewritten. It should be expanded to include a more detailed discussion of current problems, and research gaps, especially the basic characteristics of bitumen sheets used for rainwater harvesting.
- Novelty of the study, the authors need to point out their objectives clearly and novelty of the study, which need to be stressed in the revised version. Why the authors did not test the release of VOC and PAHs from the bitumen sheets? They are more toxic to the environment.
- Critical comments should be made on the results of the cited works. For example, how do the authors make the difference from background rain pollution due to air pollution? What’s the difference between different rain periods? Why did the authors consider the seasonal difference rather than the time period between two drops of rain?
- Statistical analysis of the results is missing. How many samples are taken for the water samples?
- The discussion statements are speculations. A more detailed discussion of factors affecting the observed removal performance should be added. Make every attempt to improve the discussion by critically analyzing your findings. The sunlight is one factor, do you consider the sunlight time length?
- What’s the potential application of the results from this study? How to prevent the toxic compound from the first rain by using different materials?
- Summary and Conclusions are too long. A broader discussion of the significance and potential application of this specific study and the more concise and direct conclusion is appreciated.
Author Response
Reviewer #3
The manuscript presents a report on the change of solutes from roof bitumen sheets used for rainwater harvesting. The topic is interesting and brings some information to the readers. However, the manuscript still need comprehensive revision to improve its scientific quality, rather than a report.
- Title needs to be changed, delete “Inorganic and Organic”, because it means all compounds.
DONE
- Abstract should be rewritten. The authors need to provide the general results they obtained with direct values, for example, the concentration of the pollutants.
The results and concentrations of the different pollutants are too complicated to be generalized and explained in an abstract. Nevertheless, in the revised abstract we provided information about total EC of the first flush water, compared to non-first flush water. In addition, we emphasized at the abstract that it includes both organic and inorganic elements.
- Introduction should be rewritten. It should be expanded to include a more detailed discussion of current problems, and research gaps, especially the basic characteristics of bitumen sheets used for rainwater harvesting.
Parts of the introduction were rewritten, some references were added and more information was added.
- Novelty of the study, the authors need to point out their objectives clearly and novelty of the study, which need to be stressed in the revised version. Why the authors did not test the release of VOC and PAHs from the bitumen sheets? They are more toxic to the environment.
We hope the revised introduction better explains the novelty of the paper.
This is true that VOC and PAHs are problematic pollutants that may be emit from the bitumen sheets as detailed in the introduction. The focus of our work was not the chemical composition of the water, rather the dynamics of total solutes emission from the BS, under various environmental conditions. In the revised MS we emphasize this point, and hypothesizing that the release of these toxic pollutants is likely to be similar to our observations, yet future studies are needed.
- Critical comments should be made on the results of the cited works. For example, how do the authors make the difference from background rain pollution due to air pollution? What’s the difference between different rain periods? Why did the authors consider the seasonal difference rather than the time period between two drops of rain?
Both 'Introduction' and 'Results and discussion' sections were revised to address the three comments above.
As detailed in the MS, rain samples were collected and analyzed and it is clearly seen that the high loads of solutes are not related to the rain.
- Statistical analysis of the results is missing. How many samples are taken for the water samples?
It is detailed in the M&M section and we emphasized at the caption of figure #1 that " Each data point is of a single sample".
- The discussion statements are speculations. A more detailed discussion of factors affecting the observed removal performance should be added. Make every attempt to improve the discussion by critically analyzing your findings. The sunlight is one factor, do you consider the sunlight time length?
Another section was added to the discussion section, where the all results, from the three sections of the study, are being discussed and integrated.
- What’s the potential application of the results from this study? How to prevent the toxic compound from the first rain by using different materials?
This is being discussed at the revised summary and conclusions section. [P16, L456-467]
- Summary and Conclusions are too long. A broader discussion of the significance and potential application of this specific study and the more concise and direct conclusion is appreciated.
Done. Large parts of the summary and conclusions section were removed to the discussion section and edited.
Round 2
Reviewer 3 Report
The revised manuscript improved signifciantly. I have no further comments.